# BBREFINEMENT: AN UNIVERSAL SCHEME TO IMPROVE PRECISION OF BOX OBJECT DETECTORS

## ABSTRACT

We present a conceptually simple yet powerful and flexible scheme for refining predictions of bounding boxes. Our approach is trained standalone on GT boxes and can then be combined with an object detector to improve its predictions. The method, called BBRefinement, uses mixture data of image information and the object's class and center. Due to the transformation of the problem into a domain where BBRefinement does not care about multiscale detection, recognition of the object's class, computing confidence, or multiple detections, the training is much more effective. It results in the ability to refine even COCO's ground truth labels into a more precise form. BBRefinement improves the performance of SOTA architectures up to 2mAP points on the COCO dataset in the benchmark. The refinement process is fast; it adds 50-80ms overhead to a standard detector using RTX2080, so it can run in real-time on standard hardware. The code is available at https://gitlab.com/irafm-ai/bb-refinement.

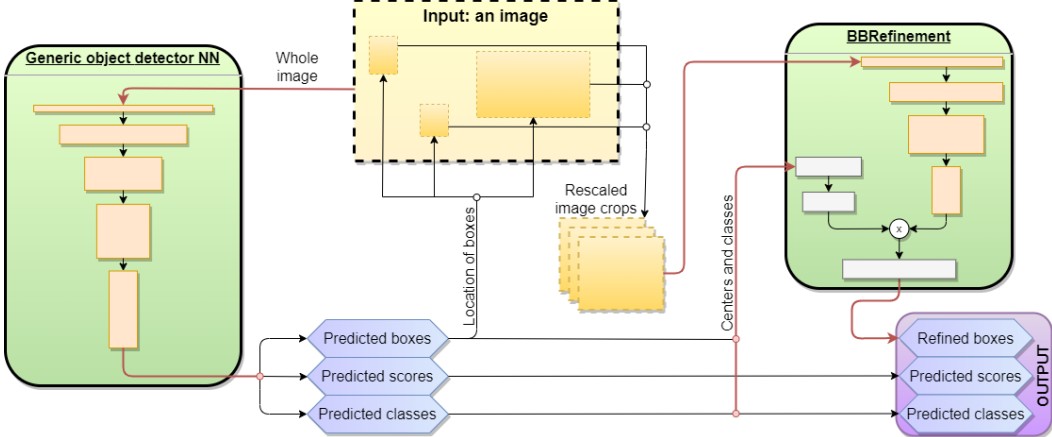

Figure 1: The figure illustrates the proposed pipeline of prediction. A generic object detector processes an image, and then the detected boxes are taken from the original image, updated by BBRefinement, and taken as the output predictions.

## 1 PROBLEM STATEMENT

**Object detection** plays an essential role in computer vision, which attracts a strong emphasis on this field among the researchers. That leads to a situation when new, more accurate, or faster object detectors replace the older ones with high frequency. A typical object detector takes an image and produces a set of rectangles, so-called bounding boxes, which define borders of objects in the image. The detection quality is measured as an overlap between the detected box and ground truth (GT), and it is essential for two reasons. Firstly, the criterion used in benchmarks – mean Average Precision (mAP) – is based on particular thresholds for various values of Intersect over Union (IoU) between the prediction and the GT. Such thresholds are typically applied to distinguish between accepted and rejected boxes in detection. Therefore, precision here is crucial to filter valid boxes

from discarded. Secondly, the more precise the detected box is, the more accurate the classification should be. Although NN-based classifiers can deal with some tolerance in shifted or cropped data, the higher accuracy in the object detection may lead to the increased accuracy in the classification process.

Existing solutions for object detection yield accuracy around 0.3–0.5mAP on the COCO dataset (Lin et al., 2014). Such a score allows the usage in many real applications. On the other hand, there is space for improvement. A combination of the following may reach such growth: more precisely distinguish between classes; increase the rate of true-positive detections; decrease false-positive detections; or increase the IoU of the detections. There are four points on why object detection may be difficult in general, which blocks further mAP growth. 1) A neural network has to find all objects in an image. The number may vary from zero to hundreds of objects. 2) A neural network has to be sensitive to all possible sizes of an object. The same object class may be tiny or occupy the whole image. 3) A network usually has no a priori information, which should make the detection easier, like the context of the scene or the number of objects. 4) There is a lack of satisfactory big datasets. Therefore, the distribution of data is sampled roughly only. In this paper, **we propose BBRefinement, which can suppress the effect of all the four mentioned difficulties.** The proposed inference scheme 'Detection $\rightarrow$ Refinement' is achieved by a combination during prediction phase with a generic detector, and it increases the IoU of the detected boxes with its ground truth labels, resulting in higher mAP.

**Related work.** The problem of refinement can be tracked to the origin of two-stage detectors, where R-CNN (Girshick et al., 2014) uses a region proposal algorithm that is used to generate a fixed number of regions. The regions are classified and by bounding box regressor refined. Faster R-CNN (Ren et al., 2015) replaces the region proposal generation algorithm with a region proposal network. The same bounding box regressor can be used iteratively to obtain more precise detections (Gidaris & Komodakis, 2015; Li et al., 2017). The effect of iterative refinement may be increased by involving LSTM module (Gong et al., 2019). The aim of refinement can also be anchors; RefineDet (Zhang et al., 2018; 2020) refines them to obtain customized anchors for each cell. Cascade R-CNN (Cai & Vasconcelos, 2018) uses a sequence of bounding box regressors to create $n$-staged object detector. In Cascade R-CNN, network head $h_0$ takes proposals from the region proposal network and feeds the regressed bounding boxes to network head $h_1$ and so on. All the heads work over the same features extracted from a backbone network. The cascade scheme shows that $h_1$ is dependant on the quality of $h_0$ head. If $h_0$ includes some bias, $h_1$ balances it. Therefore, all the heads have to be trained together (part by part), and if $h_0$ is retrained, $h_1$ should be retrained as well. In contrast, BBRefinement is a trained standalone, and it is not dependent on the quality of the object detector with whom it is coupled during inference. That makes BBRefinement universal and able to be applied on various image detectors without retraining a detector or BBRefinement.

## 2    EXPLAINING BBREFINEMENT

The main feature of BBRefinement is a transformation of the problem into a simpler scheme, where an NN can be trained easily. Compared with a standard object detector, BBRefinement is a specialized, one-purpose neural network working as a single object detector. It does not search for zero-to-hundreds objects, but it always detects only a single object and does not produce its confidence. It is also missing the part responsible for classification, so it does not assess the object's class. The only purpose is to take an image with a single object within a normalized scale and **generate a super-precise bounding box**. The training is realized on boxes extracted from a dataset according to ground truth labels. When BBRefinement is trained, the fixed model can be coupled with an arbitrary detector to realize the inference. Here, the feed for BBRefinement is bounding boxes in the form of image data produced by the detector.

### 2.1    PROBLEM WITH A NAIVE SINGLE OBJECT DETECTOR

Let bounding box $\boldsymbol{b}$ be given by its top-left and bottom-right coordinates $\boldsymbol{b} = (x_1, y_1, x_2, y_2)$. Further, let us suppose a color image $f : \boldsymbol{D} \subset \mathbb{N}^2 \rightarrow \boldsymbol{L} \subset \mathbb{R}^3$. Then a neural network detecting single object is generally noted as $g : f \rightarrow \boldsymbol{b}$. To train such a network, we generally minimize term $|\boldsymbol{b} - g(f)|$ or its alternatives.

The issue comes when $f$ includes two objects at once, and the network is extended to produce two bounding boxes. The network should return $\boldsymbol{b}_1$ for the first object and $\boldsymbol{b}_2$ for the second one. However, a generic solution will predict a box $0.5(\boldsymbol{b}_1 + \boldsymbol{b}_2)$ for the both cases, or generally $1/n \sum_{i=1}^{n} \boldsymbol{b}_i$ for $n$ if we consider that the boxes have the same frequency of occurrence. A naive solution is to modify the network to detect a sparse set of objects $g : f \rightarrow \boldsymbol{B}$, where $\boldsymbol{B} = \{\boldsymbol{b}_1, \boldsymbol{b}_2, \ldots, \boldsymbol{b}_n\}$ assumes boxes in a fixed order. Such a detection scheme is not possible in general without a deeper modification of an architecture leading to the presence of a grid, etc. With no guarantee of a single object's presence only, **the naive solution cannot be used**. This problem was solved later in chronological order by a sliding deformable models / window technique (Felzenszwalb et al., 2009), two-stage techniques such as (Fast/Faster) R-CNN (Ren et al., 2015), single-stage techniques as SSD (Liu et al., 2016) or YOLO (Redmon & Farhadi, 2017) and finally by anchors-free techniques that are mainly keypoint-based (Law & Deng, 2018). Every such approach affects the architecture of the neural network and is related to a specific model.

BBRefinement, as a single object detector, would suffer the shortcomings mentioned above. The reason is that even if we have an extracted object defined by its bounding box, such bounding box for a non-rectangular object will also involve some background, which may contain other objects. The presence of other objects leads to the problem described in the previous paragraph, and in conclusion, it may confuse the neural network, thus cause improper refinement. The examples from the COCO dataset are shown in Figure 2 for the easy case and in Figure 3 for the problematic case. For illustration, the COCO dataset includes 1.7M boxes from which 47% of all boxes have an intersection with a box with the same class, and 84% of boxes have an intersection with an arbitrary-class box. To solve the problem, BBRefinement uses information about the object's class and center as well.

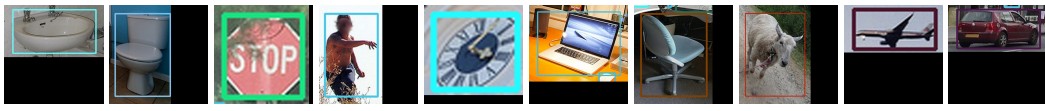

Figure 2: The figure shows crops which can be refined even with the naive way because a crop includes only one, nice visible object.

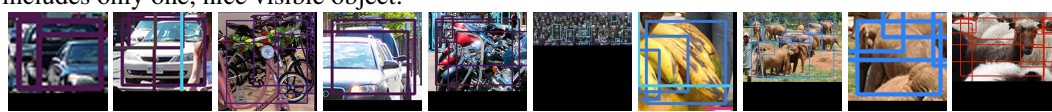

Figure 3: The figure shows crops that cannot be refined by the naive way because a crop includes multiple objects, usually of the same class. Note, precise labeling of such images is a hard task even for humans.

## 2.2 THE PRINCIPLE OF BBREFINEMENT

Firstly, we need to define a neural network in more detail, so we suppose a convolutional neural network $\mathbb{F}$ to be set of $k$ layers, $\mathbb{F} = \{f_1, f_2, \ldots, f_k\}$. Here, all the layers are meant as $f_i : D \subset \mathbb{R}^{m_i} \rightarrow L \subset \mathbb{R}^{n_i}$ which are for the sake of simplicity defined as convolutions layers without poolings/residual connections/batch norms etc, i.e., $f_i(\mathbf{M}_i) = a(\boldsymbol{W}_i \otimes \mathbf{M}_i)$, where $a$ is activation function, $\boldsymbol{W}_i$ weights and $\mathbf{M}_i$ is output of the previous layer, or, an input image in the case of $i = 1$. Such a neural network is generally called as a backbone, with the aim to map an input image iteratively into feature space. Here, we suppose $e_k : D \subset \mathbb{R}^{n_k} \rightarrow L \subset \mathbb{R}$ to be an embedding of the $k$-th layer created as $e_k(f_k(\mathbf{M}_k)) = p(f_k(\mathbf{M}_k))$, where $p$ is global average pooling or flattening operation.

Furthermore, we suppose a fully connected network $\mathbb{G}$ to be set of $j$ layers, $\mathbb{G} = \{g_1, g_2, \ldots, g_j\}$. All layers are meant as $g_i : D \subset \mathbb{R}^{q_i} \rightarrow L \subset \mathbb{R}^{e_i}$ as $g_i(\mathbf{M}_i) = a(\boldsymbol{W}_i \mathbf{M}_i)$, where $a$ is activation function, $\boldsymbol{W}_i$ weights and $\mathbf{M}_i$ is an output of the previous layer, or an input vector in the case of $i = 1$.

According to the motivation, we propose to use mixture data as an input to the suggested scheme of refinement. Convolution neural network $\mathbb{F}$ processes the input image (crop with a fixed resolution) containing an object, a fully connected neural network $\mathbb{G}$ processes a fixed-size vector that

holds information about a class and an expected center of the object. The both networks are designed in order to $|e_k(f_k(\mathbf{M}_k))| = |g_j(\mathbf{M}_j)|$ be valid. Then, both information is mixed together as $x = e_k(f_k(\mathbf{M}_k)) \cdot g_j(\mathbf{M}_j)$, where $\cdot$ is a dot product. Finally, we connect $x$ with the output layer $o$ consisting of four neurons and utilizing the sigmoid activation function. Such a neural network is trained in a full end-to-end supervised scheme. From a practical point of view, we can use an arbitrary SOTA backbone such as ResNeSt, ResNeXt, or EfficientNet, to mention a few. For BBRefinement, we use EfficientNet (Tan & Le, 2019) due to its easy scalability. In the benchmark section, we are presenting results for versions B0–B4. According to the version, an input image's resolution is $224^2$, $240^2$, $260^2$, $300^2$, and $380^2$. The version affects $|e_k(f_k(\mathbf{M}_k)|$ as well, it is 1280 (B0 and B1), 1408, 1536, and 1792.

The pipeline for the prediction with BBRefinement is illustrated in Figure 1 and is as follows. The boxes detected by a generic object detector are taken (with small padding) from the input image and rescaled into the BBRefinement input resolution. That has several beneficial consequences. Firstly, bigger objects are downscaled, and smaller are upscaled to fit the resolution, so all the objects have the same scale, which is much more effective than train a network for multiscale detection. Second, one image from the dataset yields multiple boxes. In the case of COCO, a standard detector uses 0.2M images (one image as an input), while BBRefinement uses 1.7M images (one box as an input). Also, a standard detector rescales the input images into a specific resolution to fit GPU memory, so many pixels are thrown out. BBrefinement does not use non-object parts from the image, but it allows us to use more pixels from the object-are due to a weaker downsample. Third, thanks to mixture data usage, BBRefinement obtains information about the object's detected class and center. Although such data may be imprecise, it is a piece of priory information, making the task more accessible. Finally, there is a guarantee that each crop includes just one main object. That explains why BBRefinement can produce more precise coordinates of bounding boxes than a general object detector. Note, it is necessary to take into account BBRefinement is placed on top of an object detector, which may be imprecise. As a result, the data fed into BBRefinement may be ambiguous. Therefore, the crops should not be extracted precisely but surrounded by padding; during the training, the padding is random. For the same reason, it is beneficial to distort the center by random shifts for training. Such augmentation is visualized in Figure 4.

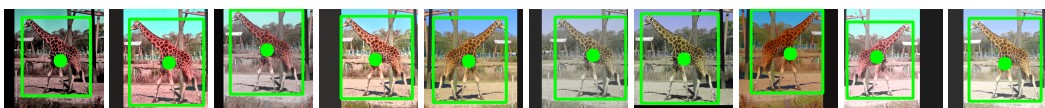

Figure 4: Augmentations of a box. The green box represents the GT label. The original crop is randomly padded. The center's position is slightly distorted (and visualized as a green dot) as we suppose BBRefinement will be applied to a generic detector's predictions, which can produce such distortion.

To realize the refinement, there are several options on **how to define loss function** $\ell$ used for training BBRefinement. The first option is to compare each coordinate of the box with the GT label using, e.g., binary cross-entropy (BCE). The second option is to use BCE for comparing top-left points and then euclidean distance for evaluating the width and height of a box. Such an approach is used, e.g., in YOLO. The third way is to use coordinates of all points to determine the boxes' areas and compute IoU. Let us imagine a situation where a box is shifted by a pixel according to its label in a vertical and horizontal direction. If we shift the box in the directions separately, we will observe that the sum of such partial losses will be equal to the loss produced by shifting in both directions at once. That is not valid behavior; the second loss should be bigger. On the other hand, the triangle inequality is fulfilled when IoU is used. The next problematic situation is when the euclidean distance is used: bigger boxes tend to produce bigger differences than small boxes. This means they produce bigger losses, and a neural network tends to focus on them more than on the small boxes. IoU is computed as a relative value; therefore, the same difference in width or height creates a bigger loss for small boxes, which is the desired behavior. Based on these reasons, we use IoU-based loss for training BBRefinement. We have two available options on how to define the IoU loss function. Namely $\ell_1(\boldsymbol{b}, \boldsymbol{b}') = -log(i(\boldsymbol{b}, \boldsymbol{b}')/u(\boldsymbol{b}, \boldsymbol{b}'))$ for the logarithmized form and $\ell_2(\boldsymbol{b}, \boldsymbol{b}') = 1.0 - i(\boldsymbol{b}, \boldsymbol{b}')/u(\boldsymbol{b}, \boldsymbol{b}')$ for the linear form, where $i$ represents intersection of two boxes, $u$ their union, $\boldsymbol{b}$ is GT box, and $\boldsymbol{b}'$ predicted box.

**Logarithmized form**. Let us consider a task where the evaluation criterion involves IoU with some threshold, such as 0.5, which can be found in many real competition websites such as Kaggle or Signate. Here, it is much more to satisfy the threshold rather than to reach the best possible IoU. Here, the goal of the BBRefinement is to take unprecise boxes (with IoU<0.5) and push them over the threshold. The logarithm-shape of IoU loss generates the biggest loss for unprecise boxes, while the loss is vanishing for high-precision boxes, similar to Focal loss (Lin et al., 2017). That is also beneficial if we take into account that no dataset is perfect, and labels created by humans are not accurate. Here, the property of smaller loss for near-perfect detections would be beneficial.

**Linear form**. The COCO dataset's official benchmark computes mAP for several IoU thresholds, such as 0.5, ..., 0.95. Here, the situation is the opposite: by refinement, pushing a box from, e.g., 0.4 into 0.95, is more beneficial than refine three boxes from 0.4 to 0.54. The reason is that a box with IoU >0.95 will be taken into account in all the IoU thresholds, while a box with IoU 0.54 will be taken into account only for the threshold of 0.5; other thresholds will count it as a false positive. Therefore, we use the linear form in the following benchmarks.

Note, both forms of the loss function can be based on a more efficient version of IoU. The other choices may be Generalized IoU loss (Rezatofighi et al., 2019), Complete IoU, or Distance IoU (Zheng et al., 2020). Generally, these IoUs converge faster and are able to compute loss effectively, even for boxes without overlap.

## 3 BENCHMARK

**The setting of the training:** BBRefinement was trained using two different computers with cards RTX 2060 or 2080. The resolution of models corresponds to the default setting of EfficientNet (Tan & Le, 2019) version, namely side size of 224, 240, 260, 300, and 380px for version B0-B4 with the batch size of 7-40 according to the version and memory of the used graphic card. For training, we used COCO dataset (Lin et al., 2014) as follows. We merged train 2014, train 2017, and a part of valid 2014. The unused part (5000 images) of valid 2014 has been used as the valid set. The testing set is represented by valid 2017. The loss function is linear IoU described above, optimizer AdaDelta (Zeiler, 2012) with default learning rate, i.e., $\alpha = 1.0$, and functionality of decrease learning rate by factor 0.5 with patience equal to two. We also experimented with cyclic LR (Smith, 2017), which converged faster but generally produced significantly worse the best loss than the used scenario. During one epoch, all training images were processed, and a single random box has been taken from each one of them. Each such box was augmented by random padding (each side separately), by linear/non-linear HSV distortion, CLAHE, and by flipping. The information about the box center has been augmented by distorting the coordinates. The illustration of the augmented box is shown in Figure 4. Models were trained until loss did not stop decrease, which took approx 70-90 epochs. For illustration, the heaviest used backbone, EfficientNet B4, was trained for nine days on a computer with an RTX2080Ti. For the comparison, we selected SOTA networks, namely Faster R-CNN (Ren et al., 2015), RetinaNet (Lin et al., 2017) (both for two various backbones), and Cascade R-CNN Cai & Vasconcelos (2018). All of them are available through Detectron2 framework[1]. Next, we used DETR (Carion et al., 2020), which is available through official implementation[2] derived from MMDetection framework. We used the reference models trained on the COCO dataset and realized the inference only.

**The detailed results** are presented in Table 1. We want to emphasize that BBRefinement improves the mAP of all but Cascade R-CNN models, considering the standard [IoU=0.50:0.95] setting, while holds that the heavier backbone of BBRefinement usually brings a stronger boost. Also, it holds that the worse the baseline model, the bigger increase of mAP. Considering the objects' size according to the COCO tools (small-medium-big), the situation is not so straightforward. In the case of small objects, EfficientNet-B1 can be marked as the best backbone with the claim that it may be beneficial to refine only the less precise models; otherwise, BBRefinement may even decrease the performance. For medium objects, BBRefinement EfficientNet-B2 is the best one, and the usage of refinement leads to an increase of accuracy in all but Cascade R-CNN models. A similar situation is for the large objects where BBRefinement EfficientNet-B4 is the best one in all the cases. There is a hypothesis that strong upscaling of small objects leads to a distortion and, therefore, to decreased performance.

---

[1]https://github.com/facebookresearch/detectron2
[2]https://colab.research.google.com/github/facebookresearch/detr/blob/colab/notebooks/detr_demo.ipynb

Table 1: mAP [IoU=0.50:0.95] performance of original and refined predictions on the COCO dataset. The table shows accuracy in the form of IOU of a generic detector when its official, pre-trained model is used – marked as a baseline. The right part shows IOU accuracy when the same pre-trained model is coupled with BBRefinement. All BBRefinement versions are trained only once, and the same trained version is used for all multiple detectors. For the training of BBRefinement, we use the same split as it is common and has also been used by the authors of the generic detectors.

| Model | Baseline | BBRefinement, EfficientNet | | | | | Boost |
| | | B0 | B1 | B2 | B3 | B4 | |
|---|---|---|---|---|---|---|---|
| All objects | | | | | | | |
| Faster R-CNN, ResNet-50 C4 1x | 35.7 | 37.4 | 37.7 | 37.9 | **38.0** | 37.9 | +2.3 |
| Faster R-CNN, ResNeXt-101 FPN 3x | 43.0 | 43.1 | 43.3 | 43.6 | **43.6** | **43.6** | +0.6 |
| RetinaNet, ResNet-50 FPN 1x | 37.4 | 38.3 | 38.6 | **38.8** | **38.8** | **38.8** | +1.4 |
| RetinaNet, ResNet-101 FPN 3x | 40.4 | 40.6 | 40.8 | **41.1** | **41.1** | **41.1** | +0.7 |
| DETR, ResNet-50 | 34.3 | 35.6 | 35.8 | **36.0** | **36.0** | **36.0** | +1.7 |
| Cascade R-CNN, ResNet-50 FPN 3x | **44.3** | 43.0 | 43.2 | 43.4 | 43.4 | 43.5 | –0.8 |
| Small objects | | | | | | | |
| Faster R-CNN, ResNet-50 C4 1x | 19.2 | 19.3 | **19.5** | 19.1 | 19.1 | 19.2 | +0.3 |
| Faster R-CNN, ResNeXt-101 FPN 3x | **27.2** | 25.8 | 25.9 | 25.9 | 25.7 | 25.9 | –1.3 |
| RetinaNet, ResNet-50 FPN 1x | **23.1** | 22.1 | 22.1 | 22.1 | 22.0 | 22.0 | –1.0 |
| RetinaNet, ResNet-101 FPN 3x | **24.0** | 23.5 | 23.4 | 23.4 | 23.4 | 23.4 | –0.5 |
| DETR, ResNet-50 | 14.3 | 15.9 | **16.0** | 15.9 | 15.9 | 15.7 | +1.7 |
| Cascade R-CNN, ResNet-50 FPN 3x | **26.6** | 24.6 | 24.5 | 24.3 | 24.4 | 24.4 | –2.0 |
| Medium objects | | | | | | | |
| Faster R-CNN, ResNet-50 C4 1x | 40.9 | 42.4 | 42.6 | 43.0 | 43.0 | **43.1** | +2.2 |
| Faster R-CNN, ResNeXt-101 FPN 3x | 46.1 | 46.6 | 46.8 | **47.2** | 47.0 | **47.2** | +1.1 |
| RetinaNet, ResNet-50 FPN 1x | 41.6 | 42.9 | 43.1 | **43.6** | 43.4 | 43.4 | +2.0 |
| RetinaNet, ResNet-101 FPN 3x | 44.3 | 44.8 | 45.1 | **45.5** | 45.4 | 45.4 | +1.2 |
| DETR, ResNet-50 | 36.6 | 38.4 | 38.7 | **38.9** | 38.8 | **38.9** | +2.3 |
| Cascade R-CNN, ResNet-50 FPN 3x | **47.7** | 46.7 | 47.0 | 47.3 | 47.2 | 47.4 | –0.3 |
| Large objects | | | | | | | |
| Faster R-CNN, ResNet-50 C4 1x | 48.7 | 52.5 | 53.2 | 53.5 | 53.5 | **53.6** | +4.9 |
| Faster R-CNN, ResNeXt-101 FPN 3x | 54.9 | 56.6 | 57.1 | 57.3 | 57.5 | **57.7** | +2.8 |
| RetinaNet, ResNet-50 FPN 1x | 48.3 | 50.7 | 51.0 | 51.3 | 51.3 | **51.4** | +3.0 |
| RetinaNet, ResNet-101 FPN 3x | 52.2 | 53.8 | 54.2 | 54.4 | 54.4 | **54.5** | +2.3 |
| DETR, ResNet-50 | 51.5 | 52.0 | 52.3 | 52.7 | **52.8** | **52.8** | +1.3 |
| Cascade R-CNN, ResNet-50 FPN 3x | 57.7 | 57.3 | 57.8 | 58.0 | 58.1 | **58.5** | +0.8 |

## 4 DISCUSSION

**Bugs in a dataset**: Deep learning, as a data-driven approach, is directly dependent on the quality of data. On the other hand, it is impossible to create a flawless dataset. The object detection task's general issues are incorrect classes, imprecise box boundaries, and missing boxes. BBRefinement is (as standard object detectors) vulnerable to the first two issues, but (opposite to standard object detectors) resistant to the third issue. If we consider missing labels as illustrated in Figure 5, we will penalize a detector during training if the detector will produce predictions for such missing labels. That will lead to decreased performance. In the case of BBRefinement, the training data are created from the labels. If some label is missing, a cropped image will not be produced. So, the missing labels only decrease the training set's size but do not affect BBRefinement's performance.

**Refinement of a dataset:** We also tested the most precise object detector, DetectoRS (Qiao et al., 2020), which can reach the mAP above 0.5. In that case, we observed a decrease of mAP by 1.3 after the refinement. We analyzed visual outputs and recognized interesting behavior: DetectoRS's predictions are closer to GT, but the refined predictions look visually better, even better than the GT. Therefore, we realized a second experiment. We took GT test labels, refined them, and visualized both of them into an image. Surprisingly, we can claim that BBRefinement can produce more precise labels than COCO dataset. On the other hand, because the boxes' positions are not identical, refined

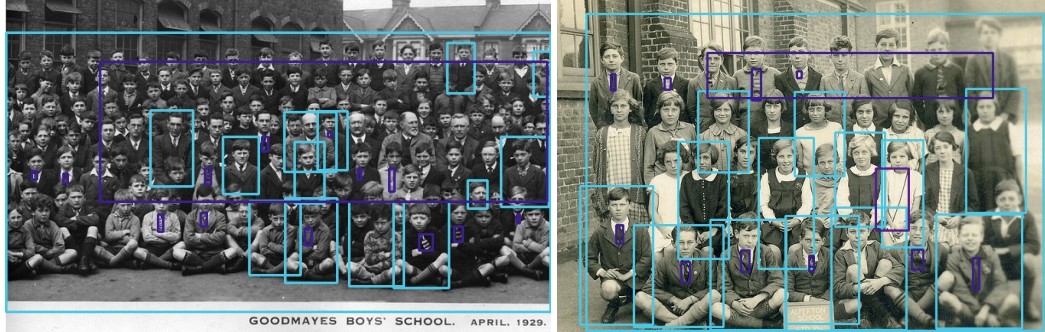

Figure 5: The figure illustrates two images taken from the COCO dataset, where the boxes are inpainted ground truth labels. It is evident that some labels are imprecise, and a lot of labels are missing. Such behavior can be seen mainly in images that include groups, and it is a known issue of the COCO dataset.

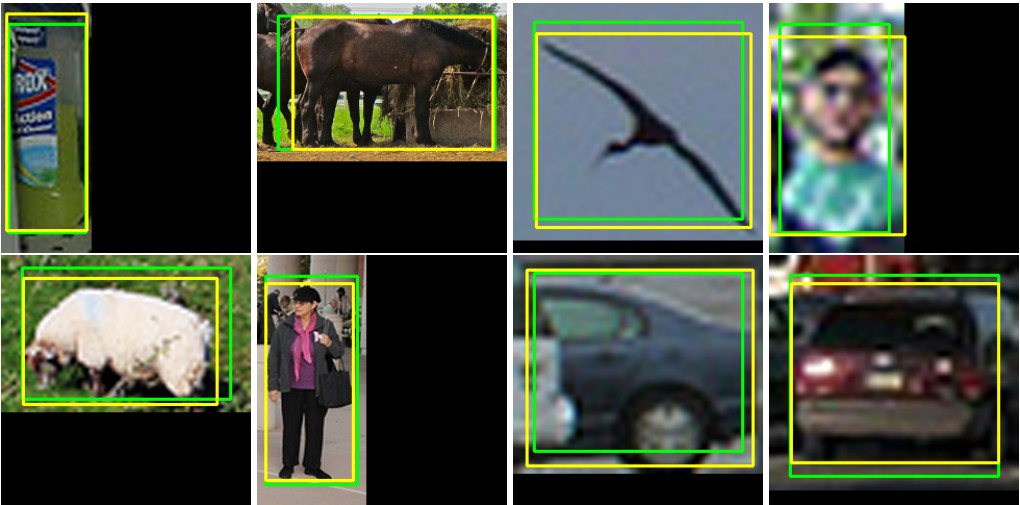

Figure 6: The image illustrated crops with green ground truth and yellow refined inpainted labels. Here, BBRefinement creates labels with significantly higher precision than is the ground truth. Best see zoomed-in.

boxes do not yield to IoU of 1.0, and therefore, the mAP can be decreased when BBRefinement is applied on a high-precise object detector. The reason why an object detector can produce predictions on a test set closer to GT than the refined version is unclear for us. Figure 6 shows crops from the test set with inpainted boxes: green color marks GT boxes given by the COCO dataset, and yellow the labels produced by BBRefinement. We selected the images in Figure 6 as such cases, where it is obvious that BBRefinement yields more precise boxes. Note, IoU between predictions and GT varies here around 0.8. Because the dataset is big, and eight selected crops were chosen selectively, we also selected eight additional crops as follows. The first one has index 100 in the ordered list of images, the second one 200, the third one 300, etc., so the selection is not affected by our preference. They are illustrated in Figure 7. We can proudly claim that BBRefinement, although not so significant as for the previous cases, still produces more precise boxes than GT (see best zoomed-in). Also, we applied BBRefinement trained on COCO to the Cityscapes dataset. Again, BBRefinement makes visually more precise labels than the Cityscapes ground truth is. Such a finding leads us to three conclusions. First, it is ambiguous to compare high-mAP object detectors because the high mAP does not necessarily mark a better detector in the meaning of real-world truth as the labels are affected by human subjection and error. Next, thanks to the high number of boxes, BBRefinement can be trained in such a generalized manner that the labeling error can vanish, so it can be used for re-labeling a dataset. Finally, there is a hypothesis that IoU between BBRefinement trained on a

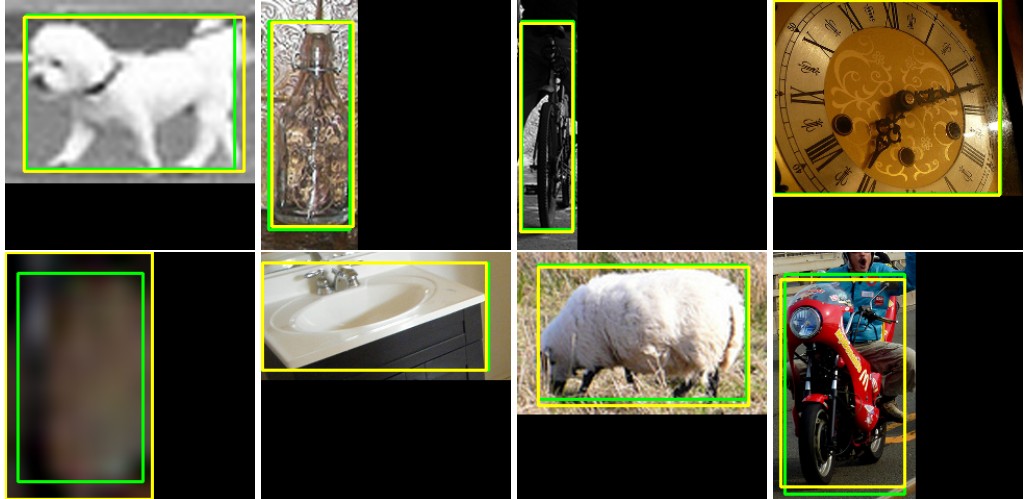

Figure 7: The image illustrated crops with green ground truth and yellow refined inpainted labels. The crops were selected uniformly according to their index to show general cases. Here, BBRefinement proposes slightly higher precision than the ground truth has. Best see zoomed-in.

specific dataset, and its GT labels can be used to express the quality of labels. The verification of this hypothesis is a theme for future work.

## 5 ABLATION STUDY

**Comparing with a naive refinement without mixture data** We have trained BBRefinement with EfficientNetB1 with the same setting as is used in Section 3, but without mixture data. It means that only visual information represented by image crops is available during training and inference. During the inference, we coupled it with 'Faster R-CNN, ResNet-50 C4 1x'. It achieved mAP performance on all/small/medium/big areas of 34.0/15.1/38.9/48.5. That is better than the baseline performance, 33.1/15.0/38.0/46.3, but worse than the full couple with the mixture data, which yields 34.9/15.5/39.5/50.2. This finding confirms the meaningfulness of the proposed scheme.

**Influence of the accuracy of center and class**: The performance of BBRefinement is affected by the accuracy of the used object detector. Therefore, we realized an experiment where GT data were distorted, fed into BBRefinement, and IOU between the refined and GT was measured. In the ideal case, IOU would be 1.0. As we show in Figure 8, we distorted the position of the center and the correct class separately. The distortion for center $c$ is realized as $c = (c_x + d_x, c_y + d_y)$, where $d_x, d_y \sim \mathcal{U}(-d, d)$ and by $d$ we mean the maximum distortion. For the class distortion, we replace $n\%$

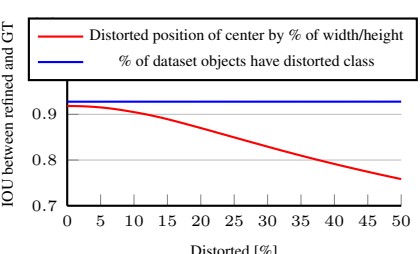

Figure 8: Influence of distortions

of correct classes with random incorrect classes. The figure shows that BBRefinement is robust on incorrect class, but it is sensitive to center position distortion. Such strong robustness in the distortion of class may, on the other hand, mean that information about the class is not important for BBRefinement, and therefore, BBRefinement can be trained even without it.

**Speed of the inference**: In some applications, the ability to run in real-time may have the same importance as high precision. Therefore, BBRefinement should not increase the inference time significantly. Via a selection of BBRefinement's backbone, the tradeoff between speed and precision can be controlled. The BBRefinement's inference time is consisting of two parts, the preparation of the crops and the inference itself. While the first part depends on the CPU, the second part relies on GPU power only. Speaking in numbers, we measured the time of BBRefinement with backbone EffnetB1 and EffnetB3. For both cases, the non-optimized preparation of crops on CPU costs 32ms

per image, where the image can include multiple crops. The time on GPU is 23ms for B1 and 44ms for B3. We could predict all crops from a single image in one batch, which helped keep the time small. The time means that BBRefinement runs 18FPS for the B1 backbone and 13FPS for the B3 one. The speed can be further increased by parallelizing crops' preparation and optimizing the model's speed by automatic tools.

**Influence of crops size** We have selected BBRefinement with EfficientNetB2 backbone and trained it for various crops size to reveal the impact. To converge faster, we have weakened the setting compared to the 'full experiment', namely 3000 steps per epoch, batch size 8, and patience 1 in reducing the learning rate. During the inference, it was coupled with 'Faster R-CNN, ResNet-50 C4 1x'. According to the graph in Figure 9, we can conclude the experiment the model is stable, and the crop

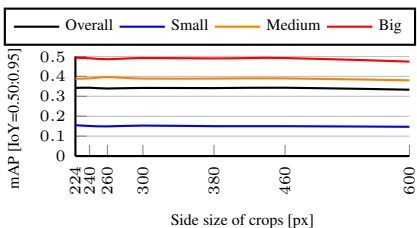

Figure 9: Influence of crops size

size has a minor impact for all sizes but 600, where is a decrease in performance. For the extreme case of crops size 600, almost all objects, including the big ones, are upsampled, which distorts them.

## 6 CONCLUDING REMARKS

We discussed the difficulties of the object detection problem. We shown the difficulties can be suppressed by the refinement stage coupled with an object detector, if the refinement is given as a single-object detector. To solve the problem when one bounding box includes more objects, we propose to use mixture data where the image information is complemented with information about the object's class and center, which helps the network to refine the desired object. We showed the simple scheme could increase the mAP of the SOTA models. Finally, we presented that our scheme, BBRefinement, is able to produce predictions that are more precise than ground truth labels.

As the refinement process is partially independent of the detector, this approach opens a new research direction. The original research, which is focused on increasing accuracy by proposing new architectures, etc., can now be complemented with independent research of refinement networks. The final system, which can be deployed to real productions on various competitions (such as Kaggle or Signate), may consist of a combination of the best algorithms from both types of research.

ACKNOWLEDGMENTS

The work is supported by ERDF/ESF "Centre for the development of Artificial Intelligence Methods for the Automotive Industry of the region" (No. CZ.02.1.01/0.0/0.0/17_049/0008414)

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
