# OpenReview forum: "BBRefinement: an universal scheme to improve precision of box object detectors"
_ICLR.cc/2021/Conference — Reject_

### Official Review · AnonReviewer4 · 2020-10-27

**Rating:** 2
**Confidence:** 5

**Review:**

##########################################################################

Summary:

This paper proposes BBRefinement, which is a post-processing for object detection to refine the predicted bounding boxes. BBRefinement takes cropped images from predicted bounding boxes as input and refine the bounding box with a separate network that is only targeted in predicting box offsets.

##########################################################################

Pros:

The proposed method is simple and experiment shows the effectiveness of proposed method.

##########################################################################

Cons:

1. Missing literature review. This paper is not the first one to study how to refine bounding boxes. There are a lot of works on refining bounding boxes [A, B, and many more], but this paper fails to discuss related works and explain the connection as well as the differences with them.

2. Missing ablation studies. Section 2.2 discusses principals of BBRefinement, including the importance of using mixture data. However, there is no experiment supporting this claim. Also the expanding ratio of bounding boxes is an important parameter, but there is also no experiment on this parameter.

3. This paper only applies the proposed method to very simple baselines like Faster R-CNN and RetinaNet. These methods (Faster R-CNN, RetinaNet) are known to predict not tight bounding boxes. I wonder if BBRefinement is still necessary when a method already predicts tight bounding boxes like Cascade R-CNN [C].

4. It is not clear to me how the model is trained, especially how boxes are sampled during training. Is it an image-centric sampling or an instance-centric sampling? Does sampling strategy matter? There could also be false positives in the prediction, does these boxes harm the training (e.g. existence of background box)?

5. Table 1 needs more explanation, do you need to train separate BBRefinement for each model? If not, what boxes do you use to train the BBRefinement.

6. The timing of BBRefinement, is "23ms for B1 and 44ms for B3" a single box? What is the average end-to-end runtime on the whole dataset? The timing is also much faster than EfficientNet speed in [D], [D] reports 52 ms for B1 and 114 ms for B3. Can you explain why your timing is 2x faster?

7. The dataset split on COCO also has problem, COCO train 2014 and part of val 2014 is exactly the same as train 2017. And the 5000 minival 2014 is exactly the same as val 2017.

##########################################################################

Reasons for score:

Although this paper presents a simple and effective solution, the overall quality of the paper is poor. First, this paper does not have discussion on related works **AT ALL**. Second, it misses important implementation details and important ablation studies. Third, this paper only applies the method to weak detectors and fails to apply it to methods that give tight bounding boxes like Cascade R-CNN. Finally, the authors put a [link to the code](https://gitlab.com/irafm-ai/bb-refinement) which leaks the authorship (one author's name and institute); this is a violation of the double-blind review policy. Considering all this facts, this paper is a clear reject to me.

##########################################################################

References:

[A] Object detection via a multi-region & semantic segmentation-aware CNN model, ICCV 2015
[B] A MultiPath Network for Object Detection, 2016
[C] Cascade R-CNN: Delving into High Quality Object Detection, CVPR 2018
[D] Designing Network Design Spaces, CVPR 2020

---

> ### Author Response · Authors · 2020-11-13
> **Official response**
>
> Dear reviewer,
>
> Thank you for your constructive comments.
>
> 1) Based on your comment, we added "Relative work," where we described similar approaches and explain why they are different. All of them are somehow integrated and trained with an existing NNs, so they cannot be definitely marked as universal. To our best knowledge, there is not such refinement that would be trained standalone on the dataset's GT labels and then coupled with an arbitrary object detector.
>
> 2) Thank you for your note. We added an ablation study section and where show the impact of the important parameters.
>
> 3) The claim "This paper only applies the proposed method to very simple baselines like Faster R-CNN and RetinaNet" is not fully correct. We also present detailed results for DETR, a totally fresh approach which is based on the Transformer. In the discussion, we also mention and present brief results of DetectorRS, which was the most precise detector at the time of writing the paper. Based on your comments, we also realized the benchmark with Cascade R-CNN and added it to the paper. The answer to the question is no, BBRefinement cannot improve the predictions taken from Cascade R-CNN. In more detail, we are able to improve the performance of large objects, but we are suffering mainly for small objects.
>
>
> 4) BBrefinement is trained separately to a standard detector. For training, we extract crops of the instances from the dataset according to GT labels, save them separately and train BBRefinement on them. So, it is instance-centric. We added the note also to the new version of the paper. So, there should be no false positives because the labels for BBRefinement are not taken as predictions from some detectors but from GT.
>
> 5)  Thank you for the note; we explained it in more detail in the new version. BBRefinement is trained only once and can be applied to a generic detector producing bounding boxes. That is also the difference between it and Cascade R-CNN. Cascade uses several networks that are connected and trained at once. So in the cascade, head 2 is trained to refine hed 1. If head 1 produces a bias, head 2 is trained to fix the bias. BBRefinement is trained using the original data, and it is just a super-precise bounding box refinement. The advantage is also easier (faster) training, application to many existing models, and faster prediction. Cascade is really heavy, while BBrefinement is a small network. The first part of the table shows accuracy in the form of IOU of a generic detector when its official, pre-trained model is used, marked as a baseline. The right part shows IOU accuracy when the same pre-trained model is used, and our schema refines the obtained results. So the generic detector is not trained by us. BBRefinement is trained once and then applied to all these models. Yes, it is super-simple, which we think is a huge benefit and yield a nice boost.
>
> 6) The time is for processing all boxes from a single image, i.e., one to tenths of boxes. All the boxes for a single image are predicted at once, in one big batch. The time expresses the time needed by the model to process it; it does not include time for extracting boxes from an image. The time (23ms for B1 and 44ms for B3) of the prediction is taken by processing all images from the dataset and by computing the mean value. The time for extracting crops is 33ms.
> The second part of the comment: I do not have an explanation of why [D] has worse values. I feel really a pity that you are referencing such a paper with prediction time, where the experiment is not fully described. The table with the time (Table 4) does not describe the conditions of the measurement. The previous table uses the setting given as "64 images on an NVIDIA V100 GPU" without explanation if the 64 images are predicted one-by-one or in one batch. We use the known implementation of EfficientNet https://github.com/qubvel/efficientnet and run it on RTX2080, no V100, P100, Titan, RTX3090 etc.
>
> 7) Thank you for the note; it is new information for us. Luckily, we did not use the minVal. Anyway, we took only the 2017 train and val and retrained for it BBRefinement with backbone B0, B1, and B3. We have realized that the new results can be marked as identical; only minor differences due to the variability in the training process are included. However, due to the long training time, we cannot retrain all the models, so we left the paper's original setting. But, the claim that we did not overfit the test dataset can be proved by Figures 6 and 7, where you can observe that BBRefiner produces different bounding boxes that the dataset has. Numerically, our predictions are worse, but visually, we produce more precise predictions.

---

> > ### Author Response · Authors · 2020-11-22
> > **continuation**
> >
> > *************************
> > I hope such an explanation helps you to understand the paper. The final application is simple but with high versatility and a nice boost in IOU. We hope the idea worth publishing because it is really helpful.
> >
> > *****************
> > The name in the repo: yes, that is a really fail:( But, no one knows the name is real or it is a name of a co-author of the paper or just of a guy who was asked to place the codes to the repository to hide the names of the real authors:) Anyway, next time, we will do it in a better way, sorry for that.

---

### Official Review · AnonReviewer3 · 2020-10-28
**Very similar to prior works**

**Rating:** 4
**Confidence:** 5

**Review:**

This paper proposes an approach, BBRefinement, to refine the bounding box predictions of an object detector. After the object detector predicts boxes, a patch is cropped around each box and passed into a separate network for refinement. The authors show that this improves the performance of several object detectors on the COCO dataset.

I think the novelty of this paper is limited as a similar idea has already been published in Cascade R-CNN [a]. The authors do not compare their work to Cascade R-CNN. To me, the major difference between this and Cascade R-CNN seems to be that Cascade R-CNN uses the RoI-pooled features for refinement, but this uses the image crops, which does not seem to be significant. Other differences such as BBRefinement use the centers and the classes of the boxes as additional information also seem to be minor. Furthermore, Cascade R-CNN seems to be able to provide better improvement on the APs. Since the authors do not provide any comparison with Cascade R-CNN, it is unclear whether BBRefinement is better at refining boxes than Cascade R-CNN or significantly different.

The authors claim an advantage of BBRefinement is that it can be trained with more images as it takes crops around each box as input while conventional detector uses the full images as input. I find this argument a little bit weird. Considering the number of actual pixels, BBRefinement actually has much fewer training data in terms of the number of pixels as large number of pixels are thrown away during cropping.

The authors also claim that BBRefinement is more robust to missing labels. The conventional detector may get penalized incorrectly for producing boxes for missing labels, while BBRefinement does not suffer from such issue as it only takes cropped images around each label as input. I don’t think this claim is fair. The authors should not compare BBRefinement to a detector because they are different. BBRefinement only refines the predictions from a detector but a detector classifies and localizes objects from an image. And actually, the regressors in convetional detectors also only process cropped images/feature maps. I am not seeing why this is an advantage to BBRefinement.

---

> ### Author Response · Authors · 2020-11-16
> **Official response**
>
> Dear reviewer,
>
> Thank you for your time and valuable comments. Here, we want to answer them.
>
> Similarity with Cascade R-CNN: in the cascade, all heads that realize bounding box regression use the same backbone convolutions. Also, each head is trained to imbalance biases and to increase precision from the previous BB regressor. Therefore, the whole process is compact, leading to high IOU, but it cannot be marked as universal. No one cannot use, e.g., trained head 2, and use it in a different model. BBRefinement is trained as a standalone regressor. So, it is trained once (on GT dataset's data) and then coupled with an arbitrary detector to increase its performance. That is the main difference, and novelty, the universality. With that, we propose a universal way to improve many existing detectors' precision, not only one. We think that is really great. But, to realize that, mixture data are necessary. The updated version of the paper includes an ablation study where the benchmark supports the claim. The new version also includes related work, including Cascade R-CNN and the explanation about the difference and novelty that we write above.
>
> The number of images/pixels: you are right that our crops that are training data do not include background, so the number of pixels may seem to be lesser. Let us show you the next example. We suppose we have a COCO dataset. As the detector, we suppose RetinaNet. According to the RetinaNet paper, the images will be resized/downscaled into constant resolution, mainly due to fit limited GPU memory (we do not consider 8xV100 computer, etc., only standard cards), so the resolution can be, e.g., 600x600px in a letterbox format, which is standard. Furthermore, let us suppose BBRefiner with EfficientNetB3 backbone, where the input resolution is 300x300px. It is obvious that if the input image includes more than four boxes, BBRefinement will have more pixels available. Note, COCO has, on average more than eight boxes per image. We have added a short explanation of it also into the new version of the paper.
>
> Robustness against missing labels: you are right, the claim is unfair, but it is true. It is right that "The regressors in conventional detectors also only process cropped images/feature maps", but some two-stage NNs are trained end-to-end, so the full output is the aim of optimization. If an image has a missing label, then such trained two-stage NN (including BB regressor part) is penalized. If there is a missing label in our scheme, then the particular crop is not used, and, therefore, BBRefinement is not penalized.
>
> *************
> We hope our explanations make it more clear, and you will possibly update your evaluation.

---

### Official Review · AnonReviewer2 · 2020-10-28
**Bad submission on object detection**

**Rating:** 2
**Confidence:** 5

**Review:**

-The idea of this paper is just to crop the detections and then forwarded to a second stage for more accurate predictions. This idea can be traced back to the original R-CNN paper, which is not even referred and discussed. There are also many papers having a second stage to refine the detection predictions, e.g. Cascade R-CNN, RefineDet, Revisiting RCNN, but none of them are discussed in this paper.

-The writing is terrible.

-Tons of literature is missing.

-To be honest, this paper should be desk rejected.

=====updates========

After reviewing the other reviews and rebuttal, I will remain my original recommendation.

---

> ### Author Response · Authors · 2020-11-10
> **This review should be desk rejected;)**
>
> Unfortunately, as you do not follow the reviewer rules and your remarks are neither comprehensive nor with recommendations, it is hard to take something helpful from your review.
>
> We agree that more literature can be added to the paper; thank you for this feedback. The updated version of the paper describes related work in more detail and also discussed the differences.  We also realized experiments using Cascade R-CNN, as is indirectly suggested by you (and directly by other reviewers). However, we can't work with "writing is terrible" (grammar? math? chapter ordering?) or "this paper should be desk rejected", as they are not bringing any information value for us.
>
> Nevertheless, thank you for your time devoted to reading our contribution.
>
> Best regards, authors.

---

### Official Review · AnonReviewer5 · 2020-11-06
**simple method to refine detection results.**

**Rating:** 4
**Confidence:** 4

**Review:**

Summary:
This paper presents a simple yet powerful and flexible framework to refine the predictions of a two-stage detector. The approach can produce more precise predictions by using mixture data of image information and the objects' class and center. They showed a simple scheme can increase the mAP of the SOTA models and it is able to produce predictions that are more precise than ground truth.

Weakness:
+ The idea of this paper is to use a refinement module to boost the performance of the two-stage detectors. I find this work to contain very limited novelty that other researchers can use/build on. The proposed method simply uses a naive refine module to extract the Region feature from the crop. In my opinion, this simple module is similar to Cascade R-CNN. The only difference is that it extracts features from the crop of the images. It does not advance the understanding of this field although is reasonable to me.

+ The ablation experiments are weak and inadequate. Only one experiment is provided to compare the performance of the refine module. The author should do more ablation studies to support his contribution. e.g. (1) the comparison with the Cascade R-CNN which extracts the feature from the region feature maps rather than the images. (2) the architecture or the refinement module number.

+ The claim of run in real-time on standard hardware, without any time cost or FPS results in this paper.  However, the speed of the refinement module may be slow owing to the extracting feature from the crops. For the two-stage detector, e.g. FPN, the proposals of the detector are 512 under the common setting.

+ The results would have been more complete if results were shown in a setting where the region feature is used without the use of the original crops. In other words, an ablation study on the effect of the feature extraction strategies.

+ How important is the crop size to the proposed method? Considering the paper states that this is required to get a good crop, some ablation studies on showing the crop strategies would be useful for understanding.

+ In Abstract, the author of this paper provides his code which is non-anonymous. It shows that the repository of BBREFINEMENT is the "IRAFM AI" and the author's name is easy to be found.  This behavior violates the rules of the anonymous code mentioned in the Author Guide.

Finally, I suggest rejecting the paper. BTW, The author should pay attention to the rules the next time.

---

> ### Author Response · Authors · 2020-11-13
> **official answer**
>
> Thank you for your review and the time. Unfortunately, it seems you did not fully catch the idea. Based on it, we are now working on a new version of the paper, where we will be more explicit. In the meantime, we will answer the comments:
>
> 1) We are not boosting two-stage detectors' performance only; we have in the benchmark also RetinaNet or DETR. Also, it is not true that there is no novelty, and it is similar to Cascade R-CNN. BBRefinement is trained only once, separately, and can be applied to a generic detector producing bounding boxes. It is combined with a generic detector during the prediction phase only. That is also the difference between it and Cascade R-CNN. Cascade uses several networks that are connected and trained at once. So in cascade, head 2 is trained to refine head 1. If head 1 produces a bias, head two is trained to fix the bias. BBRefinement is trained on original data, and it is just a super-precise bounding box refinement due to the usage of the mixture data (the impact is now proved in ablation study). The advantage is also easier (faster) training, application to many existing models, and faster prediction. The novelty is such standalone training with the usage of mixture data and combination during inference. We did not find such an approach in the literature so far. We think we are really novel with such separate refinement, which, according to our results, leads to a universal usage and a nice boost of mAP.
>
> 2) We agree; we created a section ablation study, which now has four parts.
>
> 3) It is not true we do not mention time cost. There is explicitly written in the paper, " For both cases, the not-optimized preparation of crops on CPU costs 32ms per image, where the image can include multiple crops. The time on GPU is 23ms for B1 and 44ms for B3. We could predict all crops from a single image in one batch, which helped keep the time small. The time means that BBRefinement runs 18FPS for the B1 backbone and 13FPS for the B3 one. The speed can be further increased by parallelizing crops' preparation and optimizing the model's speed by automatic tools." I feel pity that you missed it. Anyway, we add more details about the time measurements to the updated version of the paper.
>
> 4) We cannot involve various feature extraction features. There is a necessity to use original image information. Without that, it would not be able to be used with an arbitrary detector. Let us recall, BBRefinement is trained separately and such one trained model can be used for an arbitrary detector, which is also already trained and produce bounding boxes.
>
> 5) We added the experiment to the ablation study. It showed that the model is stable, and the size of crops does not have any impact until a too big resolution is chosen.
>
> 6) Yes, that is a really fail:( But, no one knows if the name is real or it is a name of a co-author of the paper or just of a guy who was asked to place the codes to the repository to hide the names of the real authors:) Anyway, next time, we will do it in a better way, sorry for that.
>
> *********
> We feel really sad that rejection is also given by the fact that the idea was not fully understood. I hope that our answers and the new version of the paper will help to make it more clear.

---

### Author Response · Authors · 2020-11-22
**Official global response**

We would like to thanks the reviewers for their time and evaluation. We are convinced they helped to improve the level of the paper. Each of the comments is accessed separately with a complete response. In general, we updated the original paper with the new version where the following changes have been made.

* We added information to the abstract that the BBRefinement is trained in a standalone manner and combined with a generic detector in the prediction phase. We also added information about precise timing.

* We added related work to the "Problem Statement" section, where we recall the previous similar works and explain the difference between existing standard refinements and our universal refinement.

* We created a more exhaustive description of Table 1.

* We added Cascade R-CNN to the benchmark.

* We add the section Ablation study. It includes four points now:
** A comparison of BBRefinement performance with a naive refinement without mixture data. It shows that the usage of mixture data is necessary.
** The influence of the accuracy of center and class. It was also included in the previous version.
** Influence of crops' size.
** Speed of the inference. It was also included in the previous version, but the current version describes the setting of the measurements in more detail.

* We added information about the number of pixels in images, downscaled images, and crops. Based on that, we can claim that BBRefinement can utilize more information than standard object detectors.

* Generally, the paper has been processed, and a lot of changes have been made to make it more clear.

---

### Decision · Program_Chairs · 2021-01-07
**Final Decision**

**Decision:**

Reject

**Comment:**

The reviewers appreciate the simplicity of the approach, but found the exposition lacking. There were also concerns about strong similarities to CascadeRCNN, which were not resolved in the rebuttal.
In the end all reviewers recommend rejection. The AC sees no reason to overturn this recommendation.

---

> ### Comment · ~Petr_Hurtik1 · 2021-01-15
> **author's response**
>
> We accept the decision. But, the claim 'strong similarities to CascadeRCNN, which were not resolved in the rebuttal' is not true. We exhaustively explained the difference in the comments below as well as in the paper.